



# Mapping global forest age from forest inventories, biomass and climate data

Simon Besnard[1,2], Sujan Koirala[1], Maurizio Santoro[5], Ulrich Weber[1], Jacob Nelson[1], Jonas Gütter[1,4], Bruno Herault[6,7], Justin Kassi[8], Anny N'Guessan[8], Christopher Neigh[9], Benjamin Poulter[9], Tao Zhang[10,11], Nuno Carvalhais[1,3]

[1]Max Planck Institute for Biogeochemistry, Germany

[2]Laboratory of Geo-Information Science and Remote Sensing, Wageningen University & Research, The Netherlands

[3]Departamento de Ciências e Engenharia do Ambiente, DCEA, Faculdade de Ciências e Tecnologia, FCT, Universidade Nova de Lisboa, Portugal

[4]DLR, Institute of Data Science Data Management and Analysis, Germany

[5]Gamma Remote Sensing, Switzerland

[6]INP-HB, Institut National Polytechnique Félix Houphouët-Boigny, Côte d'Ivoire

[7]Cirad, University of Montpellier, UR Forests & Societies, France

[8]Université Félix Houphouët-Boigny, UFR Biosciences, Laboratoire de Botanique, Côte d'Ivoire

[9]NASA Goddard Space Flight Center, Biospheric Sciences Lab., Greenbelt, MD, USA

[10]University of Florida, Department of Biology, United States

[11]University of Minnesota, Department of Forest Resources, United States

*Correspondence to*: Simon Besnard (sbesnard@bgc-jena.mpg.de) and Nuno Carvalhais (ncarvalhais@bgc-jena.mpg.de)





**Abstract.** Forest age can determine the capacity of a forest to uptake carbon from the atmosphere. Yet, a lack of global diagnostics that reflect the forest stage and associated disturbance regimes hampers the quantification of age-related differences in forest carbon dynamics. In this study, we provide a new global distribution of forest age circa 2010, estimated using a machine learning approach trained with more than 40,000plots using forest inventory, biomass and climate data. First, evaluation against the plot level forest age measurements reveals that the data-driven method has a relatively good predictive capacity of classifying old-growth vs. non-old-growth (precision = 0.81 and 0.99 for old-growth and non-old-growth, respectively) forests and estimating corresponding forest ages (NSE = 0.6 and RMSE = 50 years). Yet, there are systematic biases with overestimation in young and underestimation in old forest stands, respectively. Globally, we find a large variability of forest age with the old-growth forests in the tropical regions of Amazon and Congo, and young forests in China and intermediate stands in Europe. On the other hand, we find that the regions with high rates of deforestation or forest degradation (e.g., the arc of deforestation in the Amazon) are largely composed of younger stands. Assessment of forest age in the climate-space shows that the old-forests are either in cold and dry regions or in warm and wet regions, while young-intermediate forests span a large climatic gradient. Finally, a comparison between the presented forest age estimates with a series of regional products reveals differences rooted in different approaches as well as in different in-situ observations and global-scale products. Despite showing robustness in cross-validation results, additional methodological insights on further developments should as much as possible harmonize data across the different approaches. The forest age dataset presented here provides additional insights into the global distribution of forest age in support of a better understanding of the global dynamics in the forest water and carbon cycles. The forest age datasets are openly available at https://doi.org/10.17871/ForestAgeBGI.2021 (Besnard et al., 2021). For anonymous access during review, please refer to the data availability section below.

## 1 Introduction

Forests cover about 30% of the terrestrial surface of our planet and store a large part of the terrestrial carbon, indicating their fundamental role in the terrestrial carbon cycle (Bar-On, Phillips, and Milo, 2018). Yet, drivers controlling the capacity of the terrestrial biosphere to sequester carbon remain poorly characterized, limiting our understanding of the global land carbon sink's location (Cook-Patton et al., 2020). Such uncertainties on the geographical distribution of the carbon sink have been partly attributed to the fact that forest regrowth and demography are not systematically considered for understanding changes in the forest carbon sink (Pugh et al., 2019, Zscheischler et al., 2017).

While the recent increase in the forest carbon sink is controlled by environmental changes such as carbon dioxide ($CO_2$) fertilization, nitrogen deposition, and climate change (Zhu et al., 2016), the dynamics in the forest carbon balance are also attributed to disturbance history and forest regrowth (Pugh et al., 2019; Besnard et al. 2019; Amiro et al., 2010). Forest disturbances cause physical damages to vegetation properties and changes in forest demography, thereby affecting the balance of terrestrial $CO_2$ exchange with the atmosphere by temporarily increasing respiration and reducing photosynthesis (Birdsey et al., 2006; Johnson and Curtis, 2001; Liu et al., 2011; Schimel, 2007; Williams et al., 2012; Woodbury et al., 2007). The changes in the strength of carbon uptake or release can alter the forest carbon balance by converting forest ecosystems from carbon sinks to sources (Amiro et al., 2010; Bowman et al., 2009; Ciais et al., 2014; Moore et al., 2013). Odum (1969) hypothesized the first theory to describe the ecosystem development in the absence of major disturbance, suggesting that the age of forests and how demographic changes drive carbon accumulation. Yet, an intrinsic property of a stand can be modified to varying degrees of changes in environmental



conditions and disturbance, therefore slowly change along with a forest age or successional continuum (Irvine et al.,
2005; Piponiot et al., 2018).
Despite the sensitivity of the forest carbon balance to disturbance and regrowth (Buitenwerf et al., 2018; Sulla-Menashe
et al., 2018), existing empirical models and current bottom-up spatiotemporal assessment of $CO_2$ fluxes do not
explicitly account for these effects (Jung et al., 2020; Tramontana et al., 2016; Jung et al., 2011). By not explicitly
constraining data-driven statistical models with disturbance history or forest demography, the forest carbon balance in
regions with newly disturbed forests and old-growth forests may not be realistically estimated. For instance, large
discrepancies are observed between statistical bottom-up approaches (e.g., FLUXCOM initiatives,
http://www.fluxcom.org/) and atmospheric inversions in estimating net ecosystem exchange (NEE), particularly in the
tropics where site history plays a substantial role in NEE magnitude (Pugh et al., 2019). To account for the contribution
of disturbance on the land carbon sink, information on the geographical distribution of disturbance is therefore required,
albeit such information is rather limited at the global scale (Ciais et al., 2014). Forest age, related to time since
disturbance, can be seen as a useful surrogate in analyses of the impact of disturbance on the ability of forests to
sequester and store carbon. Incorporating forest age into terrestrial biosphere modelling offers a starting point to
characterize disturbance history, therefore to get more insights on the location of the terrestrial carbon sinks (Pugh et al.,
2019). Reliable estimates of gridded forest age are, therefore, an important and needed source of information. The
recent advances in describing the geographical distribution of forest demography globally (Huang et al., 2010; Kennedy
et al., 2010; Poulter et al., 2019) have paved the way to consider forest age and disturbance history in carbon cycle
studies.
Here, we aim to provide a new gridded global forest age dataset *circa* 2010 inferred from a compilation of forest
inventory, biomass and climate data. More specifically, we introduce the *in-situ* forest inventory dataset and the
modelling framework used in this study as well as the predictive capacity of the presented model to infer forest age at
the plot level. We further describe the global and regional patterns of the gridded forest age dataset and their
uncertainties. The presented forest age dataset is finally benchmarked against a series of independent regional and
global datasets.
**2 Method**
**2.1 Forest inventory and climate data**
The globally gridded forest age dataset was developed by collecting *in-situ* plot level stand age, and aboveground
biomass (AGB) estimates from a series of forest inventory databases (Álvarez-Dávila et al., 2017; Anderson-Teixeira et
al., 2018; Anderson-Teixeira et al., 2016; Baker et al., 2016; Johnson et al., 2016; Lewis et al., 2013; Mitchard et al.,
2014; N'Guessan et al., 2019; Poorter et al., 2016; Schepaschenko et al., 2017; Somogyi et al., 2008; Sullivan et al.,
2017). Besides, we sampled 20,000 observations from the US Forest Service Forest Inventory and Analysis (FIA) data
(Burill et al. 2018) containing i*n-situ* plot level stand age and aboveground biomass (AGB) estimates (the original
number of observations in the FIA dataset = 350,000). To reduce the unbalanced sample size across age classes, we
weight-sampled the FIA data with decadal age classes that are underrepresented in the dataset before including the FIA
data having higher weights. The weights for each decadal class were calculated following Eq. (1):
$$weight_i = \frac{\sum_{i=1}^{n} N\,age\,class_i}{Total\,N} \tag{1}$$
Where i is a decadal class and N is the number of observations.
The methods used in inventory surveys to estimate stand age relied on expert knowledge, tree diameter measurements,
tree rings from cores of selected trees (e.g., Burill et al. 2018), and/or semi-directive interviews (e.g., N'Guessan et al.,
2019). Forest inventory plots were classified as old-growth forests when stand age was more than or equal to 300 years.
In total, the final dataset had around 25,000 plots and around 44,000 observations. Geographical biases were observed
in the compiled dataset (Fig. 1) with North America, Europe and South East of China being well represented, while
Africa, Indonesia, and Australia being either underrepresented or not represented at all. The Amazon basin and the West
part of Eurasia were relatively well represented. Besides, stand age data were generally collected in locations easily
accessible, therefore unmanaged forests in remote areas were very likely less represented than managed forests.

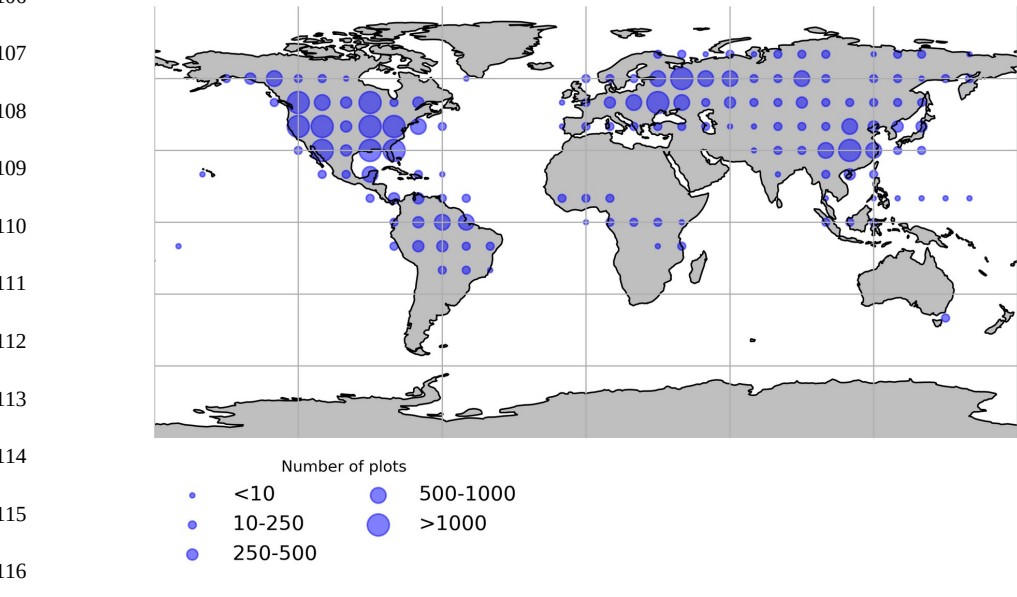

**Figure 1** Spatial distribution of the forest inventory plots used for the forest age maps. Each dot represents the total
number of plots within 10x10 degrees.
A broad meta-analysis of the compiled dataset (Fig. 2) revealed that the observations covered a large spectrum in the
climate-space (Fig. 2A), although in hot and dry regions few plots were collected probably due to the low presence of
forest ecosystems in such regions. We further described the age spectrum covered at the regional scale and found that a
large spectrum of forest age was cover in North America (Fig. 2B) and Eurasia (Fig. 2C), while in the tropics biases
were observed (i.e., large fraction of tropical old-growth forest and relatively young forests) (Fig. 2D).

131

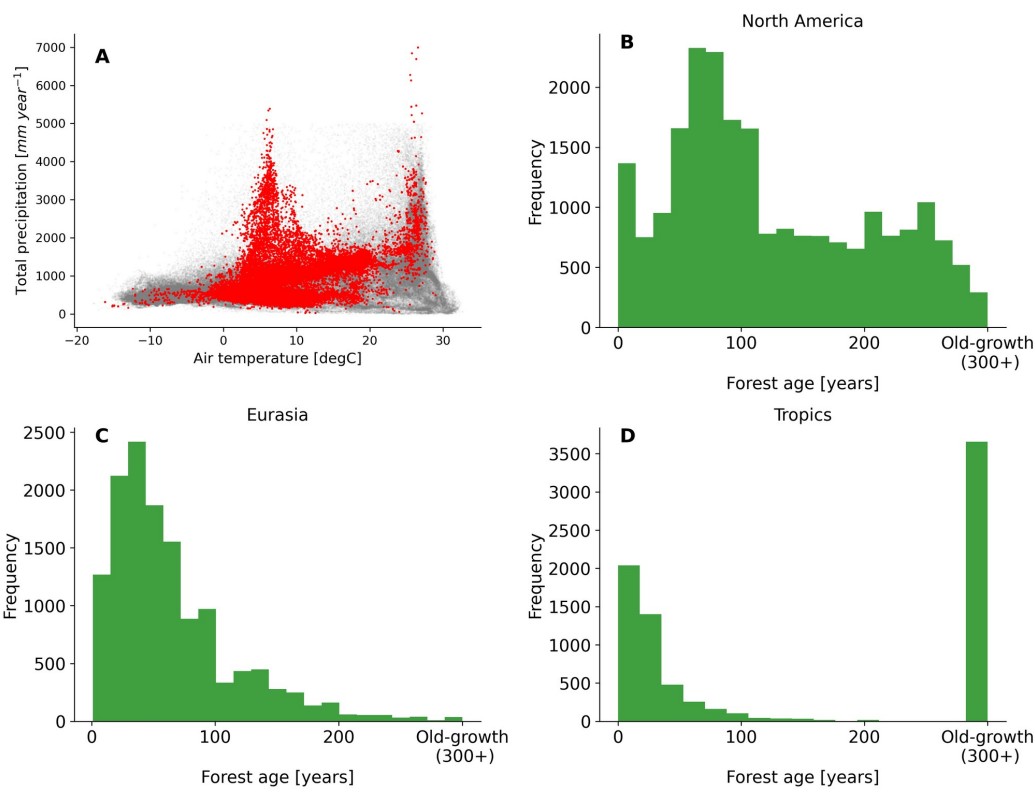

**Figure 2** Distribution of the forest inventory plots in a climate space defined by air temperature and total annual
precipitation (A). Histogram distributions of the forest age observations in North America (B), Eurasia (C) and the
tropics (D) are also shown. The grey dots show the global distribution of 0.25° grid-cell forest in climate space defined
by air temperature and precipitation, while the red dots show the distribution of the forest inventory data in the same
climate space.

For each forest inventory plot, we extracted bioclimatic variables from the WorldClim version2 (Fick and Hijmans,
2017). Table S1 summarizes the list of covariates considered in our study. Two datasets were further created. A non-old-
growth forests dataset that contained the plots with a reported stand age estimates ranging from 1 to 299 years-old and a
binary dataset reporting whether an observation had an age estimate less than 300 years-old or whether an observation
had an age estimate more than or equal to 300 years-old or not reported but considered as old-growth tropical forests
(0= non-old-growth forest and 1= old-growth forest).

**2.2 Feature selection and model training**

From the set of predictors related to vegetation and climatic conditions (Table S1), we performed a feature ranking with
recursive                        feature                        elimination                        (RFE)                        procedure
([https://scikit-learn.org/stable/modules/generated/sklearn.feature_selection.RFE.html](https://scikit-learn.org/stable/modules/generated/sklearn.feature_selection.RFE.html)) (Guyon et al., 2002) both on the



non-old-growth forest and binary datasets. The 10 best covariates selected by the RFE algorithm were further used to
train either a Random Forest (RF) regressor algorithm (RFregressor)
(https://scikit-learn.org/stable/modules/generated/sklearn.ensemble.RandomForestRegressor.html) or an RF classifier
algorithm (Rfclassifier
(https://scikit-learn.org/stable/modules/generated/sklearn.ensemble.RandomForestClassifier.html). As such, two distinct
models were implemented. The RFregressor model was used to estimate forest age in the non-old-growth forests
dataset, while the RFclassifier model was used to classify old-growth vs. non-old-growth forests using the binary
dataset (0= non-old-growth forest and 1= old-growth forest). The performances of the two models were assessed using
leave-one-cluster-out cross-validation to reduce possible spatial auto-correlation between the training and test sets
(Ploton et al., 2020). A cluster of plots contained all the plots that were within the same pair of latitude and longitude
coordinates rounded to the nearest 10th degree (e.g., latitude= 20 degrees and longitude= 110 degrees) (see Fig. 1). For
the RFregressor model, the root-mean-square error (RMSE), the normalized root-mean-square error (NRMSE) and
Nash-Sutcliffe model efficiency coefficient (NSE) were used for assessing the predictive capacity of the model for
predicting forest age. For the RFclassifier model, we reported the precision (i.e., the number of correctly-identified
members of a class divided by all the times the model predicted that class), recall (i.e., the number of members of a
class that the classifier identified correctly divided by the total number of members in that class) metrics, and F1-score
(i.e., combination of precision and recall) for assessing the predictive capacity of the classifier for distinguishing
between old-growth and non-old-growth forests. Additionally, we explored functional relationships between the
variables selected by the feature selection procedure and stand age in the RFregressor model by using the Tree SHapley
Additive exPlanations (TreeSHAP) algorithm (Lundberg and Lee, 2017; Lundberg et al., 2018). A negative SHAP value
for a given variable X translates a negative contribution to the local changes of forest age, and vice-versa.
**2.3 Upscaling procedure**
To upscale the two models (i.e., RFclassifier and RFregressor models) from plot-level data to the global scale, we
collected climate grids from the WorldClim dataset (Fick and Hijmans, 2017) and a series of AGB grids *circa* 2010 (i.e.,
corrected for tree cover with thresholds of 0%, 10%, 20% and 30%) from the Globbiomass project
(http://globbiomass.org/). The tree cover correction was done by masking-out the 100-meter pixels in the original AGB
product (i.e., 100m resolution) having tree cover estimates (Hansen et al., 2013) below one of the aforementioned tree
cover thresholds within a 1km extent. The original filtered AGB maps were further aggregated from 100m to 1km
spatial resolution using a bilinear resampling method.
The upscaling procedure was done in two steps. First, each 1km pixel was classified either as old-growth or non-old
growth forests using the trained RFclassifier model. Second, the 1km pixels classified as non-old growth were assigned
with an age estimate ranging from 0-299 years inferred from the RFregression model, while the pixels classified as old-
growth forest were assigned a default age value of 300 years. In total, four gridded forest age maps with a 1km spatial
resolution were obtained using the different aforementioned AGB maps derived from the different tree cover thresholds
(hereafter MPI-BGC forest age datasets). From the 1km resolution forest age maps, we also created maps that reflected
the fraction of several age classes (0-300+ with decadal resolution) within each 0.5-degree grid cell resolution.





**3 Results and discussion**
**3.1 Model development and evaluation**
We used the 10 most important variables from the set of variables presented in Table S1 indentified by the RFE
algorithm procedure for the RFregression and the RFclassifier models (Table 1). This set of selected variables was
further used to train the two models both in the cross-validation analysis and the global upscaling procedure.
**Table 1** List of the predictors confirmed as important by the feature selection algorithm for RFregression and the
RFclassifier models. See table S1 for details on the variable names.

| Model setup | Vegetation | Climate |
|---|---|---|
| **RFregression** | agb | Isothermality, MaxTemperatureofWarmestMonth, MeanDiurnalRange, MeanTemperatureofWettestQuarter, PrecipitationofWarmestQuarter, PrecipitationofWettestMonth, PrecipitationSeasonality, srad, vapr |
| **RFclassifier** | agb | AnnualMeanTemperature, AnnualPrecipitation,Isothermality, MeanTemperatureofColdestQuarter, MeanTemperatureofDriestQuarter, MinTemperatureofColdestMonth, TemperatureAnnualRange, TemperatureSeasonality, vapr |


By assessing the cross-validation results, we found that the RFclassifier model was able to accurately partition old-
growth and non-old-growth forests with precision estimates of 0.81 and 0.99 for old-growth forest and non-old-growth
forests, respectively (Fig. 3A). Furthermore, we found recall values of 0.94 and 0.98 for old-growth forest and non-old-
growth forests, respectively, while we found F1-scores of 0.87 and 0.99 for old-growth forest and non-old-growth
forests, respectively (Fig. 3A). The performance of the RFregression model was relatively high (NSE= 0.60, RMSE=
47.63 years and NRMSE = 51.52%) (Fig. 3B), while the model residuals across 10-degree latitudinal averages were
relatively low (Fig. 3C). However, the quantile-quantile plot depicted biases in both very young and old forests (Fig.
3D). More precisely, the RFregression model slightly overestimated the age estimates of young forests while it
underestimated the age estimates of older forests (i.e., >150 years old) at the plot level. The biases for the very young or
the older forests were probably due to either the properties of the training dataset in which older forests are still largely
underrepresented compared to younger stands (Fig. 2A-C) or the statistical method used (i.e.,Random Forests). Such
biases could potentially be propagated from plot level to global scale and have implications in representing the location
of younger and older forests globally.



Open Access · Earth System Science Data · Discussions

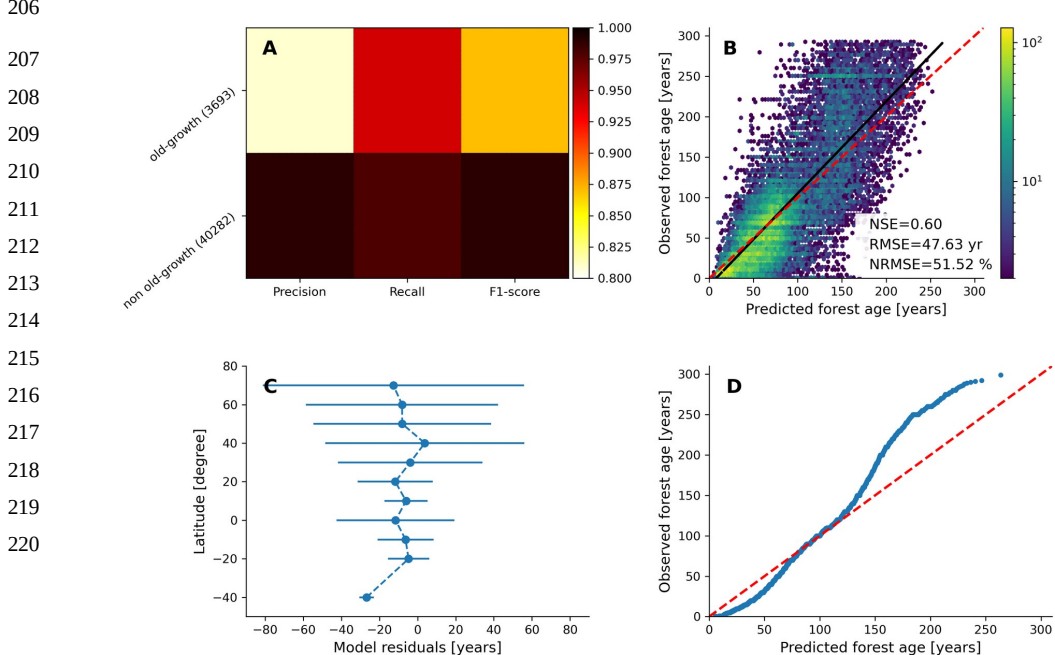

**Figure 3** Cross-validated results of the old-forests vs. non-old-forests classification (A) and comparison of predicted vs. observed forest age estimates from the regression model (B). In C, the average model residuals ∓ standard deviation within 10-degree latitudinal beans are shown. The quantile-quantile plot (D) is also shown.

We further investigated the variable importance of the selected variables and the functional relationships learned by the RFregression model between forest age and these selected variables. For this, we computed the SHAP values for each predictor to show how each predictor contributes, either positively or negatively, to the forest age estimates. First of all, we observed that vapr was the most important variable followed by agb and MeanTemperatureWettestQuarter (Fig. 4). While it was expected to have biomass (i.e., agb) as an important variable in predicting forest age, it was interesting to find that a proxy for atmospheric water demand (i.e., vapr) had a strong control on forest age. Such high importance of vapr could suggest, for instance, an association between high atmospheric demand for water (i.e., dry conditions) and disturbance intervals (e.g., fire frequency) (Mueller et al., 2020), therefore impacting the age distribution at the plot level.
















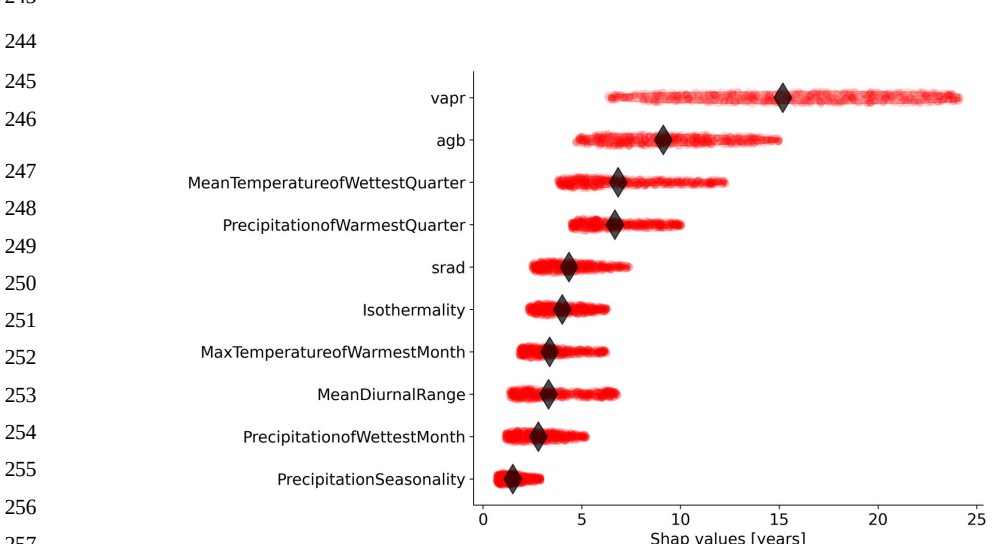

**Figure 4** Relative importance of the independent variables selected by the feature selection algorithm in predicting
forest age estimates. Each dot represents the absolute SHAP value of one observation. The diamond represents the
median value for each variable.

The emergent relationships revealed that an increase in AGB was associated with an increase in the forest age estimates
(Fig. 5A). This was expected as older trees have a higher amount of carbon stored in the aboveground components
compared to younger forests. The modelled forest age estimates appeared to be also relatively sensitive to the climatic
conditions. For instance, we observed that climatic conditions with low water atmospheric demand (i.e., low vapr) (Fig.
5C) promoted older forests as well as conditions with high solar radiation (Fig. 5E), such as in the wet tropics. Finally,
we observed that changes in forest age were also associated with air temperature conditions (Fig. 5E-G) and
precipitation regimes (Fig. 5H and Fig. 5I).



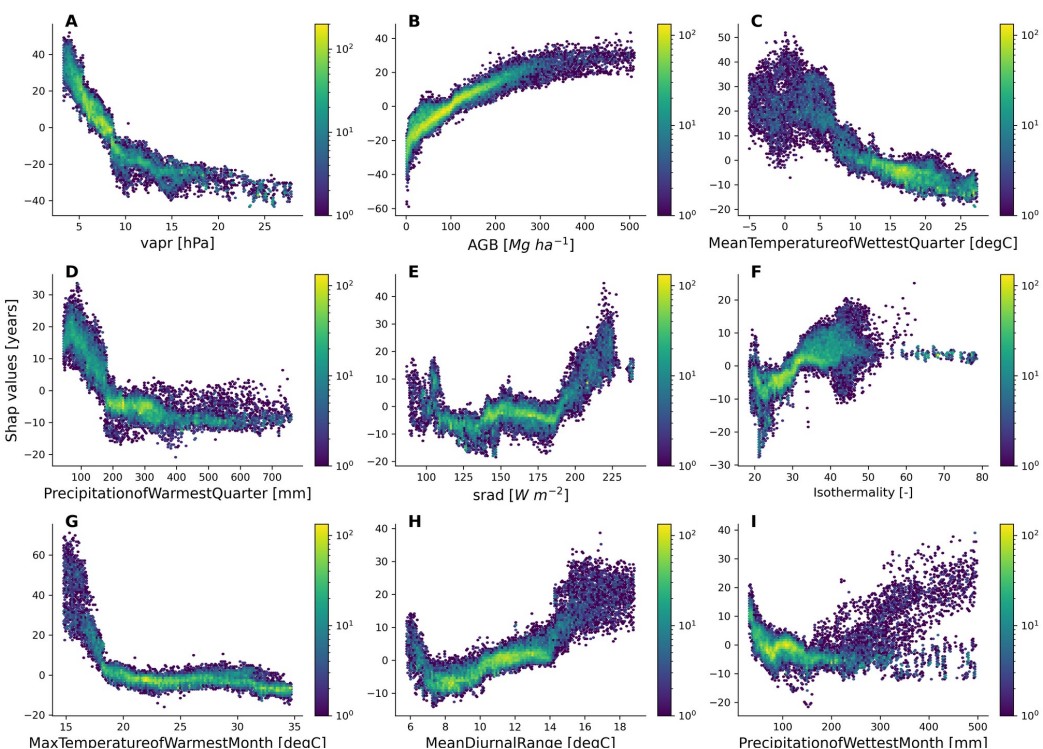

**Figure 5** Emergent relationships between the retrieved SHAP values and the independent variables selected by the feature selection algorithm.

### 3.2 Global forest age patterns and regional overview

The MPI-BGC forest age product shows a large range of forest age across the globe (Fig. 6). We observed that the most represented age class was the old-growth forests with around 1,1 billion hectares, while a limited fraction of very young forest was observed (i.e., < 10 years old) (Fig. 6A). Not surprisingly, most of the old-growth/undisturbed forests (+300 years old) can be found in the Amazon basin (Fig. 6B), the Congo basin (Fig. 6C) and part of the Indonesian peninsula (Fig. 6H), where the minimal human disturbance occurred. A large area occupied by very young forests was found in the Southeast part of China (Fig. 6H), probably due to the implementation of afforestation/reforestation policies as well as natural disturbances (Zhang et al., 2017). Similarly, young and intermediate forests were found in the African tropical dry forests (i.e., Sahel and Miombo regions) (Fig. 6C), where the frequency of the fire regimes is very high resulting in a relatively young age-class structure (Werf et al., 2017). Large scale fires in the North American boreal region also resulted in widespread patches of younger forests as well as a mosaic of stands of different ages since they last burned (Fig. 6G). On the other hand, the unmanaged part of the North American boreal region near the ecotone, where fires are more infrequent, revealed older stands (Fig. 6G). Forests in British Columbia were generally old, although patches of younger forests probably in the early stages of recovery from disturbance were also observed. European forests were in young/intermediate stages of forest succession (Fig. 6E). The increase of harvested forest area (Ceccherini et al., 2020) and considerable afforestation practices (Naudts, et al., 2016) were probably explaining a relatively young to intermediate forest demography as well as a mosaic of different age classes in the European region. The region of

Siberia revealed a gradient of younger to older forests going from the South to the North part of the Siberian region
(Fig. 6F). Such an observation could suggest different fire regimes between Southern and Northern Siberia. Finally,
Australian forests were relatively young in the North part of the country while a mosaic of age class dominated the
Southern part of Australia (Fig. 6D). The age patterns observed in the Northern part of Australia somehow correspond
to the fact forests are defined as regrowth in this region (Pugh et al., 2019). Yet, it is important to note that the few forest
inventory plots in regions such as Australia (Fig. 1) could limit our certainty on the forest age estimates attributed by the
statistical approaches due to extrapolation issues.

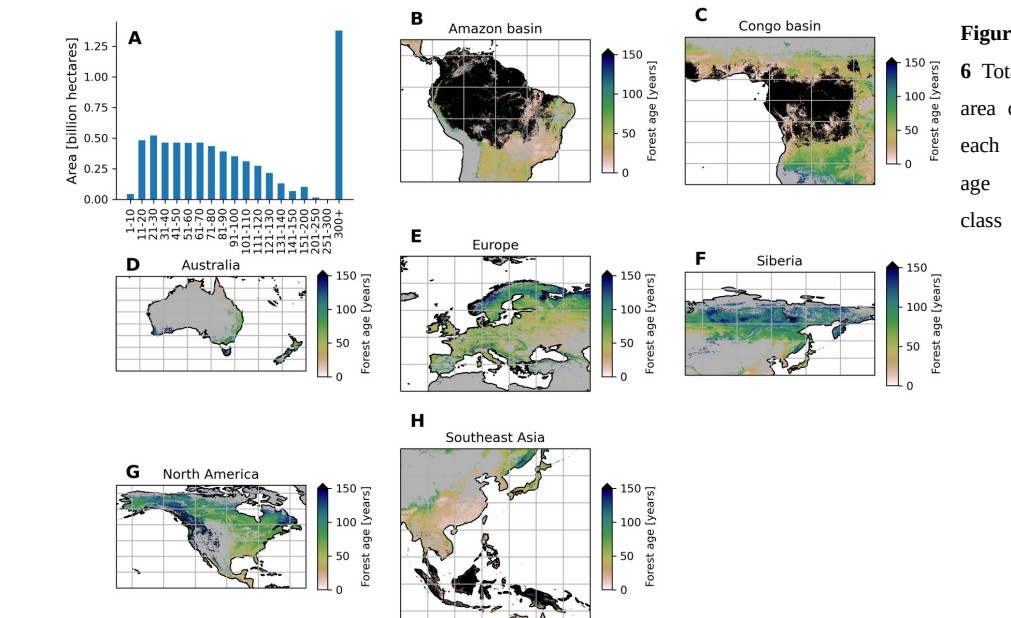

**Figure 6** Total area of each age class globally (A) and close-up examples in the Amazon Basin (B), Congo Basin (C), Australia (D), Europe (E), Siberia (F), North America (G) and Southeast Asia (H). For better visualization, the forest age estimates in the close-up examples (B-H) range from 0 to 150 years-old. The forest age map using a 10% tree cover threshold is shown.

**3.3 Global forest age relationships with atmosphere, hydrosphere and vegetation conditions**

We further investigated the distribution of the forest demography in the climate and vegetation  spaces (Fig. 7).
Generally, we observed that with warmer (i.e., air temperature) and drier (i.e., VPD) conditions, forest appeared to be
younger with the expectation of old-growth tropical forests located in relatively warm climatic conditions (Fig. 7A).
Not surprisingly, we found that most of the old-growth tropical forests were located in regions with high productivity
(i.e., high GPP and high biomass) (Fig. 7B), which coincides with our previous results investigating the structure of the
statistical model showing that an increase in forest biomass was coupled with an increase in forest age (Fig. 5A). On the
other hand, we observed that younger-intermediate forests were more productive than older forests outside the tropical
old-growth forest envelope. More precisely, for similar carbon stocks, we found that forest being less productive will
tend to belong to an older age class. Mature forests were found in cool temperatures and moderately low precipitation
conditions (Fig. 7C), where rates of fast growth but slow decomposition generally drive forest dynamics, therefore
where mature forests can potentially be found. Younger stands, on the other hand, were found in relatively warm
conditions but in a wide range of precipitation regimes (Fig. 7C). Finally, while a large fraction of young forests were
located in regions with low water availability and high water atmospheric demands, we also observed that above a
certain threshold of water availability (i.e., > 0.4-0.5), the amount of water available for trees (i.e., IWA) was not
directly associated with changes in forest age unlike VPD (Fig. 7D).

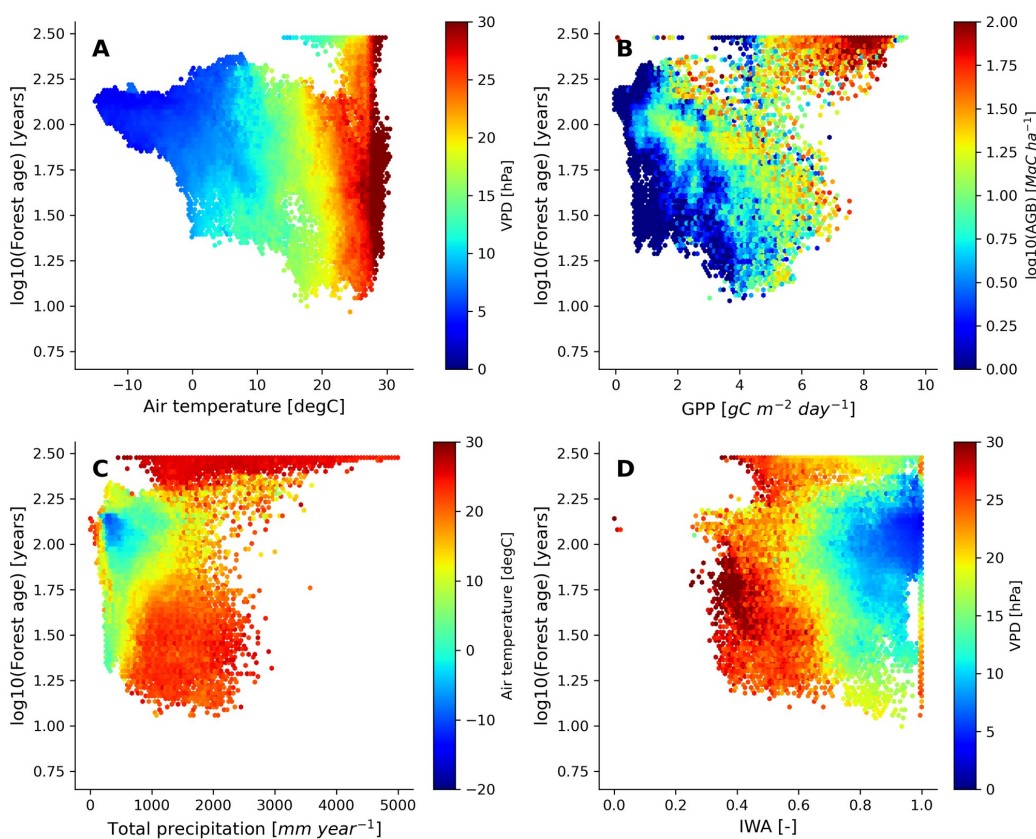

**Figure 7** Forest age distribution in the climate, hydrological and productivity spaces defined by air temperature, vapour
pressure deficit, total precipitation, soil water availability, GPP and above-ground biomass. The forest age map used
here corresponds to a tree cover threshold of 10% aggregated to 0.25 degree using a weighted average of all non-
NODATA contributing pixels. GPP is gross primary productivity derived from the FLUXCOM RS+meteo product
(Tramontana et al. 2016; Jung et al. 2011; Jung et al. 2020) and IWA is an index for soil water availability (Tramontana
et al. 2016). The climatic variables were retrieved from the ERA5-reanalysis data
(https://apps.ecmwf.int/datasets/licences/copernicus/). For all the climatic variables, we computed an annual mean for
the year 2010.
**3.4 Sensitivity analysis, uncertainties and comparison with previous products**
We performed a sensitivity analysis using a series of AGB gridded products filtered with different tree cover thresholds
to produce different global age products (see method section) (Fig. 8). This analysis showed that in South America,
mainly the dry regions were sensitive to the tree cover threshold being applied, with forest age estimates being lower



when no tree cover threshold was applied compared to a 30% tree cover correction (Fig. 8A). Similarly, we observed that the dry parts of the Congo basin depicted a sensitivity to the applied tree cover thresholds (Fig. 8B). In Europe, we observed widespread differences between the forest age estimated without a tree cover correction and with a tree cover correction (Fig. 8C). Generally, forest age estimates were higher when the 30% tree cover correction was applied. In Siberia (Fig. 8D), North America (Fig. 8E) and Southeast Asia (Fig. 8F), there were also large patches of forest where correcting the biomass maps with a tree cover threshold led to substantial differences in the age estimates. Overall, such observations were expected as mosaic vegetation, due to management practices or disturbance regimes, in the dry tropics (forest/grassland/shrubland), in Europe (forests/croplands) and Northeast of the United States (forests/croplands) are largely represented within a 1km grid cell, which could explain the sensitivity of the forest age estimates to tree cover thresholds in these regions.

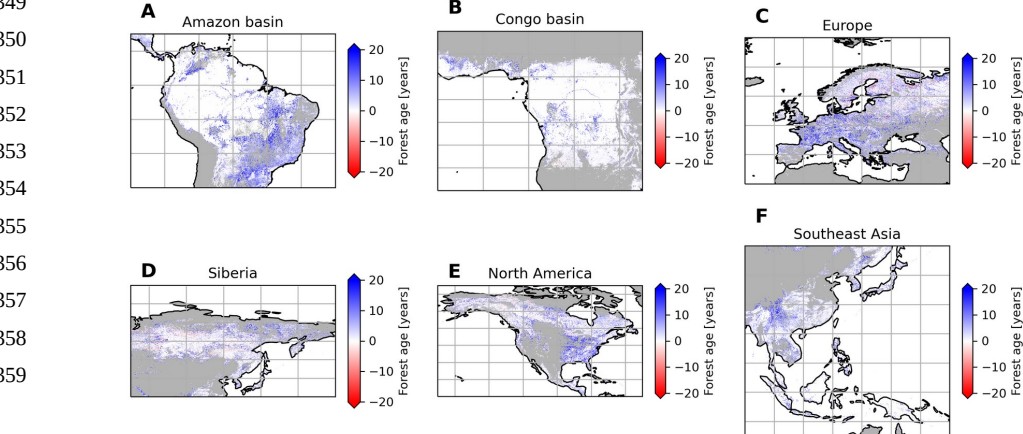

**Figure 8** Sensitivity of the presented age product using 30% tree cover correction thresholds or no tree cover correction. The differences between the age estimates derived from a forest biomass product using a 30% tree cover correction and the age estimates derived from a forest biomass product not using a tree cover correction are shown. Blue colour means that the age estimates are higher with the 30% tree cover correction than without correction, while the red colour means that the age estimates are lower with the 30% tree cover correction than without correction.

Besides, we explored uncertainties associated with the two statistical models used for the upscaling procedure (Fig. S1, Fig. S2 and Fig. S3). First, we observed that the RFclassifier model had overall very high probabilities to classify a non-old-growth forest pixel when being classified as a non-old-growth forest (Fig. S1), and vice-versa (Fig. S2), suggesting relatively high confidence in the partitioning between old-growth and non-old-growth forests in the MPI-BGC forest age product. The regions at the edge of the Amazon and the Congo basins appeared to have the lowest confidence in classifying old-growth vs. non-old-growth forests (Fig. S1A, Fig. S1B and Fig. S2) with a probability close to 0.5. On the other hand, we observed relatively high probabilities for classifying non-old-growth forests in Europe (Fig. S1C), Siberia (Fig. S1D), North America (Fig. S1E) and Southeast Asia (Fig. S1F). We also provided uncertainties in predicting forest age estimates by retrieving the 25%, 50%, and 75% quantile predictions from the RFregressor model for computing the inter-quantile range (IQR, quantile 75% - quantile 25%) divided by the median (i.e., quantile 50%) of the forest age estimates (IQR/median) (Fig. S3). While in Europe (Fig. S3C), China (Fig. S3F) and the Eastern United States (Fig. S3E) the IQR/median estimates were relatively low, we observed high IQR/median estimates in Northern



North American regions (Fig. S3E) as well as in large patches of Siberia (Fig. S3D) and the dry tropics (Fig. S3A and
Fig. S3F).
We further compared the spatial patterns of the MPI-BGC forest age dataset with a series of independent regional and
global forest age products (Chazdon et al., 2016; Pan et al., 2011; Poulter et al., 2019; Zhang et al., 2017) (Fig. 9 and
Fig. 10). In the Amazon basin, we found that the MPI-BGC forest age product depicted widespread higher forest age
estimates (i.e., blue colour) than the Chazdon et al. (2016) dataset (Fig. 9A), resulting in a substantially bigger area of
tropical old-growth forest in the MPI-BGC forest age product (Fig. 9B). On the other hand, we observed lower forest
age estimates in the regions of Rio Grande Do Norte and Paraiba in the MPI-BGC forest age product (i.e., red colour).
Such disagreement between the two products could be related not only to the different methods used to infer forest age
(i.e., statistical method vs. age-AGB chronosequence approach for the MPI-BGC forest age and the Chazdon products,
respectively) but also to the uncertainties of the RFclassifier for classifying old-growth vs. non-old-growth forests in
this region (Fig. S1 and Fig. S2). Similarly, the presented product and the Pan et al. (2011) dataset revealed widespread
discrepancies in the North American region, particularly in the Western part of the United States and the North
American boreal forests (Fig. 8E). More precisely, the Pan et al. (2011) dataset had a higher fraction of young forest
patches than the MPI-BGC forest age product (Fig. 9F). Methodological differences between the Pan et al. (2011) and
the MPI-BGC forest age datasets could explain such differences. While forest inventory, fire polygon data and remote
sensing and a multi-stage approach were used to retrieve forest age estimates in the Pan et al. (2011) dataset, the MPI-
BGC forest age product relied on forest inventory and climate data as well as statistical methods. Additionally, forest
inventory plots used to derive the MPI-BGC forest age product were relatively sparse in Canada (Fig. 1), which might
limit the statistical methods used for the MPI-BGC forest age product to predict realistic forest age estimates (i.e.,
extrapolation issues). Finally, the forest age estimates of the MPI-BGC forest age product in China were rather
consistent with the Zhang et al. (2017) dataset (Fig. 9C). The area distribution across age classes of the two products
appeared to have a relatively good agreement in China (Fig. 9D).

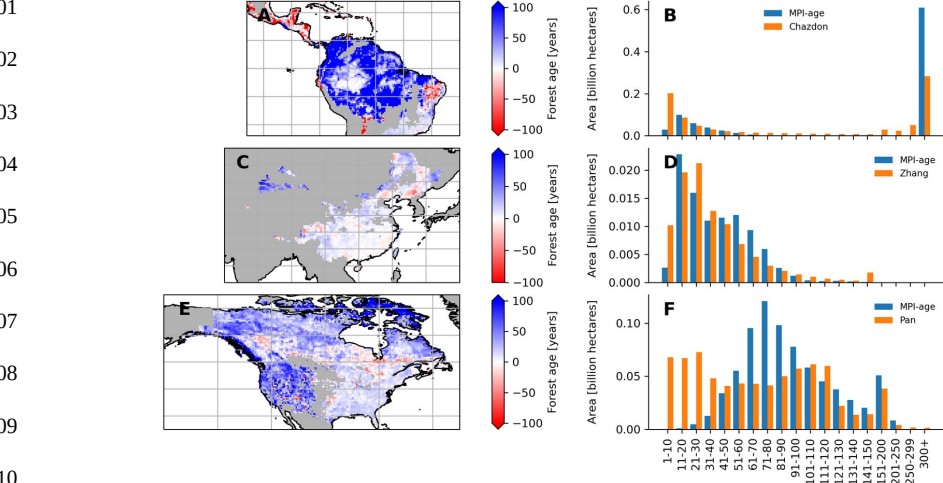

**Figure 9** Comparison between the forest age dataset from this study and independent forest age dataset: Amazon basin
(A and B), China (C and D) and North America (E and F). For a fair comparison with the independent age datasets, the
MPI-BGC forest age map used here is the one without tree cover correction applied to the AGB dataset. Differences



were computed using weighted age estimates from the fraction of the decadal age classes within each 0.5-degree grid
cell resolution.
We also found large and widespread discrepancies between the MPI-BGC forest age dataset and the global forest age
dataset (GFAD) (Poulter et al., 2019) (Fig. 10). Overall, the GFAD product had both higher fractions of very young
forests and old forests (Fig. 10B). Because the GFAD used a different AGB product for the pan-tropical region and
mainly relied on coarse national forest inventory data for the Northern hemisphere, widespread differences were
expected between the GFAD and the MPI-BGC forest age maps. For instance, the MPI-BGC forest age dataset depicted
older forests in the Western part of the United States (i.e., blue colour), while it showed younger forests across Europe
than the GFAD product (Fig. 10A). Differences were also apparent in the dry tropics, where the MPI-BGC forest age
dataset showed younger forests than the GFAD product, particularly in the Miombo regions. Such observations could be
explained by the integration of MODIS fire information in the GFAD forest age dataset. As such, we adjusted the MPI-
BGC forest age dataset with the forest age product inferred from the MCD45A1 MODerate-resolution Imaging
Spectroradiometer (MODIS) fire product at 1 km resolution (Giglio et al., 2018, Poulter et al., 2019), which was used in
the GFAD product. In this MODIS-age product, forest age was determined as the last time since a fire event occurred
within a grid cell for the period 2000-2015, thereby assuming that the entire pixel was burned down. For instance, forest
age within a 1 km grid cell was 5 years old if the last time a fire occurred within this grid cell was in 2010. When
adjusting the MPI-BGC forest age dataset with the MODIS-age product, the latter took precedence over the former
dataset. As expected, we observed a higher fraction of younger forest in the adjusted MPI-BGC forest age dataset (Fig.
S4B), although large discrepancies between the two products remained when comparing the weighted average forest
age estimates at the pixel level (Fig. S4A).

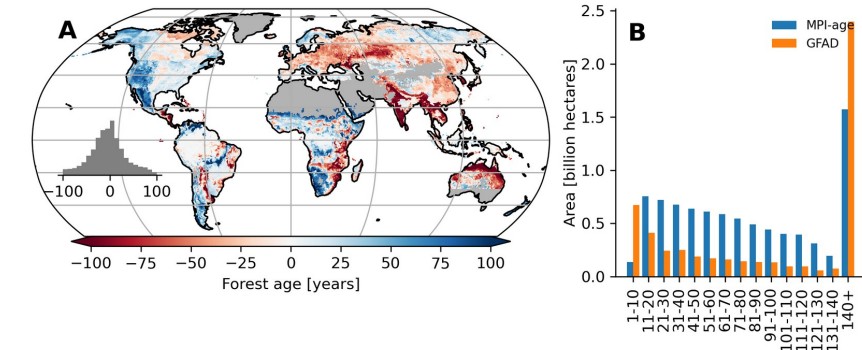











**Figure 10** Difference between the forest age estimates derived from the MPI-BGC forest age product and the GFAD
product. Differences were computed using weighted age estimates from the fraction of the decadal age classes within
each 0.5-degree grid cell resolution.
**4 Data availability**
The dataset of the different forest age products presented in this study can be downloaded from the Data Portal of the
Max Planck Institute for Biogeochemistry at https://doi.org/10.17871/ForestAgeBGI.2021 (Besnard et al., 2021). For
anonymous access during review, the data are available at https://nextcloud.bgc-jena.mpg.de/s/Xwt8AdkHkgL3TTc.




## 5 Conclusion

We presented a new forest age dataset derived from forest inventory, biomass, climate and remote sensing data. Generally, the statistical model used to create the gridded age datasets had a relatively good capacity to predict forest age estimates at plot level (precision of 0.81 and 0.99 for classifying old-growth and non-old-growth, respectively and NSE of 0.6 for predicting non-old-growth forests), while biases were observed particularly when predicting older forests. The functional relationships between biomass and forest age learned by the statistical models appeared to agree with forest age theory and the role of environmental/climate in modulating the relationship. The proposed gridded datasets allowed us to assess the global patterns of forest age and provided insights into regional forest demography. For instance, relatively young-intermediate forests were observed in Europe and China where management practices and afforestation/reforestation activities are predominant. We could also demonstrate that old forests are largely represented in very wet and warm regions as well as in very cold regions. However, the comparison of the MPI-BGC forest age product with independent forest age datasets revealed large discrepancies between them, suggesting high uncertainties in mapping forest demography globally. Overall, this forest age product provides a new source of information related to disturbance history and forest regrowth, which is key to achieve a better understanding of the location of the forest carbon sinks and sources.

**Author contributions**

SB and NC designed the study. SB conducted analysis and wrote the paper under the direction of NC. SB, NC, SK, JG, and UW collected and harmonized the forest inventories datasets. BP, BH, JK, and AN provided data for the analysis. All authors contributed to the discussions and interpretation of the results and the writing of the paper.

**Competing interests**

The authors declare that they have no conflict of interest.

**Acknowledgments**

We would like to thank all the initiatives aiming to collect forest inventory plots. We thank the members of the Biogeochemical Integration Department at the Max Planck Institute for Biogeochemistry for providing feedback on the presented results. We acknowledge support by the European Union through the BIOMASCAT (https://eo4society.esa.int/projects/biomascat/) and VERIFY (776810) (https://cordis.europa.eu/project/id/776810) projects.

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
