# Peer review of "Copernicus Publications The Innovative Open Access Publisher"

_Earth System Science Data, 2021_

## Referee Comment (RC2)

Besnard et al. present a new global forest stand age product which uses a random forest trained on forest inventory data to predict forest stand age based on a gridded global biomass product and climatological variables. Forest stand age distributions have emerged as an important determinant of estimates of forest carbon uptake, with the legacy of past disturbances, particularly anthropogenic, often contributing to a large carbon sink (Caspersen et al., 2000; Kondo et al., 2018; Pugh et al., 2019; Shevliakova et al., 2009). Whilst they only capture aspects of structure related to stand-replacing disturbances and not smaller-scale changes in structure, they are a major step forward compared to neglecting the legacy effects of past disturbance completely. To my knowledge, there has, until now, only been one stand age product available with global coverage (Poulter et al., 2019). Alternative products using complimentary methods, which can help identify and address uncertainties in the existing knowledge of global forest age structure are therefore very much needed. The manuscript is clearly written and structured. The generated maps seem to show reasonable skill, at least for middle-aged (70-130 year old) forests. However, I have several questions around the methods and concerns around the interpretation which are detailed below.

**Major comments**

The method has similarities to the biomass age curve approach (Chazdon et al., 2016; Zhang et al., 2017), but instead of using observed age curves, the random forest regression effectively uses climatological data to implicitly predict stand biomass growth curves, combining these with the biomass information to produce age. I.e. it assumes that growth and (non-stand-replacing) mortality rates can be estimated well by climatological variables alone. Given the importance of edaphic variables and management for determining these rates, I was a bit surprised not to see them included in the set of predictors. Whilst management is a tricky thing to assess globally, there are at least roughly-related variables, e.g. human development index, which could have been tested. Possibly there is substantial autocorrelation of management with climate, which may help out here in terms of the accuracy of the classification, but then this would also potentially confound the current interpretation of climatic variables relating to Fig. 5. At the very least it would be good to see some discussion of the potential influence of management, if not testing the importance of some management and soil-related variables.

As things stand, I think that the interpretation of the influence of climate variables on the age estimates (Fig. 5) has to be made in the context of these variables effectively driving an implicit stand growth model which converts the biomass data into age. The biomass product is the only predictor which contains information about the current state of the forest including the cumulative effect of its full land-use, management and disturbance history. I.e. the climatic variables are not explaining age, they are (or at least likely are) explaining how biomass relates to age. A direct interpretation of climate effects on age would in any case be flawed because of management not being considered. I think that assessing the drivers of age distributions themselves (as opposed to trying to make the best map) would be clearer using a model which only included drivers and not also the state (i.e. biomass).

The masking out of low tree cover is a nice technique to reduce the negative bias on pixel biomass, and thus stand age, caused by a mix of forest and non-forest landcovers within a 1 km$^2$ pixel. However, although the size of a stand is a loosely defined concept, it is generally

much smaller than 1 km$^2$. This leaves me wondering why the authors aggregate the biomass dataset from 100 m to 1 km? Is this to reduce noise? 100 x 100 m is already much bigger than most of the inventory plots which will underlie this product, but it would make the scale mismatch between plot-level training data and the biomass product used for extrapolation much less acute. Is a reduction in noise worth the loss of this important small-scale variation? An assumption of homogeneity within a 1 km$^2$ cell (even after masking for tree cover) would tend to reduce the extremes of low and high biomass which would have been seen in the plot-level training data. This might go some way to explaining the relative dearth of young stands in the global age product (Figs. 9 and 10).

Do all training datasets resolve up to 300 years of age (Line 100)? How did you deal with this if not? Are all the methods used to determine age likely to be accurate going back this far? Given the biases in prediction at greater than ca. 130 years (Fig., 3), and that the age of old growth forest is both perspective and biome dependent, perhaps the classification between old-growth and non-old-growth would have been more accurate if a younger age threshold was taken? Whether or not you want to test this is of course up to you (perhaps you did already?), but at least a bit more clarity and discussion would be good.

I'm struggling with the interpretation around Fig. 7, where my reading of the graphs doesn't find the same features.
- L312. My reading of this graph is that in warm regions the whole range of forest ages is found, whereas in cooler regions, only a relatively narrow range of ages exists. This itself is pretty weird, as the plot seems to suggest that cold (presumably boreal) forests only have ages of around 100 years, with no younger stands. This doesn't seem very plausible.
- L320. It seems very strange that the youngest stands should only be found in the tropics (i.e. regions of high temperature, Fig. 7c). How can you explain this result?
- L323. I don't see any association between age and water availability in Fig. 7d. The points basically seem to form a square, apart from a small indent in the upper left, which seems challenging to interpret causally.

The comparison against other age products is very nice, however, not all comparator datasets are created equally. For instance, Pan et al. (2011) is based on a spatially systematic inventory system (at least in the US). Similarly, much of the temperate and boreal data in GFAD is based on summaries of statistics from national forest inventories. Whilst these come with substantial uncertainties, I would argue that they provide a sterner evaluation of the MPI-BGC product than the comparison with products based on biomass age curves. I suggest that this is reflected in the discussion of these results and also that the comparison with GFAD provides separate histograms for regions where GFAD is based on inventories and those where it is based on the biomass-age approach.

**Other comments**

Line 59. "Yet, an intrinsic..." It's unclear (at least to me) what this sentence is saying. Please can you rephrase it?

L125. Typo, "was covered in"

L230. I don't think this result necessarily implies that vapr is a strong determinant of forest age. It can also imply that vapr is a strong control of how AGB relates to age. Given that high vapr is likely to limit both growth rates and maximum biomass, this seems to me the most likely interpretation. An influence of vapr on fire frequency, as suggested in the text, would surely primarily act through the effect of fire on AGB.

Fig. 3d. Are the under and overestimations distributed evenly geographically, or concentrated in specific regions? Providing some maps which break down the bias would be very useful for interpreting the results of the upscaled map.

L288. Ceccherini et al. mainly show increases in Sweden and Finland. These increases are disputed (Palahi et al., 2021; Wernick et al., 2021) but also only occur after 2016 and therefore are not relevant to this map dated 2010.

L290. See also (Vilén et al., 2012) for data on European age class distributions.

L292. I agree that different fire regimes seem plausible (Shorohova et al., 2011), but there is also substantial harvest identified in southern Siberia (Curtis et al., 2018), which should also play into this discussion.

Fig. 4 caption. Clarification. "... in predicting forest age estimates in the regression model."

L367-368. This sentence is really hard to follow, please can you rephrase it? I think what is being presented is the fraction of the random forest ensemble which predicts an old-growth forest for pixels which the mean of the ensemble attributes to old growth, but after reading it several times I'm not 100% sure.

L392. Are you able to speculate a bit more on the methodological differences that drive the differences of the new forest age product with Pan et al.?

Figure 9. I don't follow why the age map without using a tree cover threshold needs to be applied for consistency. Surely the tree cover threshold deemed to be most appropriate should be used for comparison. Or, perhaps better, all tree cover thresholds should be compared and the one which provides the best agreement with the independent, inventory-based, datasets (at least for regions where the inventory is systematic) might be recommended? It's really helpful that you explore this uncertainty due to tree cover and make all the maps available (thanks!), but many users will need to make a choice on a single map to use and it would be helpful to have a best recommendation to support this.

Figures 6 and 9. Is this mean or median age per pixel? If mean, how did you deal with the old-growth age class having an infinite upper age bound?

Figure 10. The integrated forest area does not appear to be the same for GFAD and MPI-BGC products. Differences also appear to be shown in regions that have no forest (high northern latitudes, interior Australia). Please could you check? Also, which tree-cover correction was used in this comparison?

**References**

Caspersen, J. P., Pacala, S. W., Jenkins, J. C., Hurtt, G. C., Moorcroft, P. R. and Birdsey, R. A.: Contributions of Land-Use History to Carbon Accumulation in U.S. Forests, Science (80-. )., 290, 1148–1151, doi:10.1126/science.290.5494.1148, 2000.

Chazdon, R. L., Broadbent, E. N., Rozendaal, D. M. A., Bongers, F., María, A., Zambrano, A., Aide, T. M., Balvanera, P., Becknell, J. M., Boukili, V., Brancalion, P. H. S., Craven, D., Almeida-cortez, J. S., Cabral, G. A. L., Jong, B. De, Denslow, J. S., Dent, D. H., Dewalt, S. J., Dupuy, J. M., Durán, S. M., Jakovac, C. C., Junqueira, A. B., Kennard, D., Letcher, S. G., Lohbeck, M., Muscarella, R., Nunes, Y. R. F., Ochoa-gaona, S., Orihuela-belmonte, E., Peña-claros, M., Ruíz, J., Saldarriaga, J. G., Sanchez-azofeifa, A., Schwartz, N. B. and Steininger, M. K.: Carbon sequestration potential of second-growth forest regeneration in the Latin American tropics, Sci. Adv., 2, e1501639, 2016.

Curtis, P. G., Slay, C. M., Harris, N. L., Tyukavina, A. and Hansen, M. C.: Classifying drivers of global forest loss, Science, 1111, 1108–1111, doi:10.1126/science.aau3445, 2018.

Kondo, M., Ichii, K., Patra, P. K., Poulter, B., Calle, L., Koven, C., Pugh, T. A. M., Kato, E., Harper, A., Zaehle, S. and Wiltshire, A.: Plant regrowth as a driver of recent enhancement of terrestrial CO2 uptake, Geophys. Res. Lett., 45(10), 4820–4830, 2018.

Palahi, M., Valbuena, R., Senf, C., N, A. and Pugh, T.: Concerns about reported harvests in European forests, Nature, 592, E15–E23, 2021.

Pan, Y., Chen, J. M., Birdsey, R., Mccullough, K., He, L. and Deng, F.: Age structure and disturbance legacy of North American forests, , 715–732, doi:10.5194/bg-8-715-2011, 2011.

Poulter, B., Aragão, L., Andela, N., Bellassen, V., Ciais, P., Kato, T., Lin, X., Nachin, B., Luyssaert, S., Pederson, N., Peylin, P., Piao, S., Pugh, T., Saatchi, S., Schepaschenko, D., Schelhaas, M. and Shivdenko, A.: The global forest age dataset and its uncertainties (GFADv1.1), , doi:doi.pangaea.de/10.1594/PANGAEA.897392, 2019.

Pugh, T. A. M., Lindeskog, M., Smith, B., Poulter, B., Arneth, A., Haverd, V. and Calle, L.: Role of forest regrowth in global carbon sink dynamics, Proc. Natl. Acad. Sci. U. S. A., 116(10), 4382–4387, doi:10.1073/pnas.1810512116., 2019.

Shevliakova, E., Pacala, S. W., Malyshev, S., Hurtt, G. C., Milly, P. C. D., Caspersen, J. P., Sentman, L. T., Fisk, J. P., Wirth, C. and Crevoisier, C.: Carbon cycling under 300 years of land use change: Importance of the secondary vegetation sink, Global Biogeochem. Cycles, 23(2), GB2022, doi:10.1029/2007GB003176, 2009.

Shorohova, E., Kneeshaw, D., Kuuluvainen, T. and Gauthier, S.: Variability and dynamics of old- growth forests in the circumboreal zone: Implications for conservation, restoration and management, Silva Fenn., 45(5), 785–806, doi:10.14214/sf.72, 2011.

Vilén, T., Gunia, K., Verkerk, P. J., Seidl, R., Schelhaas, M., Lindner, M. and Bellassen, V.: Reconstructed forest age structure in Europe 1950–2010, For. Ecol. Manage., 286, 203–218, 2012.

Wernick, I. K., Ciais, P., Fridman, J., Högberg, P., Korhonen, K. T., Nordin, A. and Kauppi, P. E.: Quantifying forest change in the European Union, Nature, 592(7856), E13–E14, doi:10.1038/s41586-021-03293-w, 2021.

Zhang, Y., Yao, Y., Wang, X., Liu, Y. and Piao, S.: Mapping spatial distribution of forest age in China, Earth Sp. Sci., 4(3), 108–116, doi:10.1002/2016EA000177, 2017.

---

## Author Response (AR1)

**Authors' Response to the Review Comments**

**Journal:** Earth System Science Data

**Manuscript #:** essd-2021-77

**Title of Paper: Mapping global forest age from forest inventories, biomass and climate data**

**Authors:** Simon Besnard, Sujan Koirala, Maurizio Santoro, Ulrich Weber, Jacob Nelson, Jonas Gütter, Bruno Herault, Justin Kassi, Anny N'Guessan, Christopher Neigh, Benjamin Poulter, Tao Zhang, Nuno Carvalhais

**Date Sent:** September 28th, 2021

We appreciate the time and efforts of the editor and referees in reviewing this manuscript and the valuable suggestions offered. In addressing all issues indicated in the review report we trust that the revised version meets the Reviewers' comments and the journal's publication requirements.

**1. Response to Comments from Reviewer 1**

**The forest age is a variable measurable in those forests subject to stand-replacing disturbances only, which are:**

- **fires (only in those forest types where fires determine the total loss of biomass, e.g conifer forests (especially in boreal climate),**
- **clear-cut (although this management option is increasingly limited to some forest types only (in particular conifer boreal forests),**
- **pest may also determine a complete loss of forest biomass,**
- **other forest types and management systems do not qualify the biomass stock with an age, and the artefact assignment of an age-value may determine biases in the derivation of other variables considered to be associated with the age e.g. biomass stock, biomass growth rate.**

**The suggested way forward is to:**

- **identify those forest types where stand-replacing disturbances occur and map those**

- **use as datasets all data-points for which the age from the last stand-replacing event has been established with certainty from the latest registered stand-replacing disturbance (e.g. not just extrapolated from the biomass stock present).**

**This means that all data points for which the age has been derived from the biomass stock level only have to be excluded from the analysis; unless have been collected in those forest types likely subject to stand-replacing disturbances (e.g. forest fire in boreal forests); which means that for most of the boreal conifer forests such age derivation from the biomass stock may be done (although for instance it cannot for boreal rainforests, unless subject to clear-cut).**

- **apply the methods described to identify the most significant variables to extrapolate age to those forest land for which age data are available**
- **assign an NA to those lands for which an age-value cannot be assigned with certainty, e.g. all rainforests not subject to clearcut.**
-

**Response:**

Thank you for all the suggestions. We agree that disturbances regimes and management practices influence the age-AGB relationship and growth/mortality rates and that it would have been relevant to add such variables as covariates. However, it is relatively challenging to infer management practices and disturbance regimes at the global level. To our knowledge, there is no global product describing management systems globally.

Yet, we derived a series of proxies for disturbance and management regimes. We intended to do so by creating two proxies for management/disturbance regimes derived from the Hansen tree cover dataset (Hansen et al., 2010, Science):

- The intensity of tree loss from the Hansen tree cover loss layer (Hansen et al., 2010, Science). This metric was derived by counting the 30m pixels that experienced a tree cover loss for the period 2000-2019 within 1km.
- Last time since tree cover loss from Hansen tree cover loss layer (Hansen et al., 2010, Science) – median and standard deviation metrics. This metric was calculated as the last time from 2019 since a 30m pixel experienced tree cover loss and we further computed the both the median and the standard deviation of this last time since tree cover loss within 1km.

These three statistical layers are now included as part of the global forest age product to allow the user to mask out pixels that did not experience disturbances in the last 20 years or the pixels that have rather hetheregenous stand age.

Nevertheless, we disagree that age cannot be determined in tropical forests that were not subject to clear cut, albeit more uncertain than recently disturbed forests. For this reason, we did not assign an NA value to the pixels for which an age-value cannot be assigned with certainty, as suggested by the referee. Instead, we let the user decide whether he/she wants to mask out pixels that did not experience disturbances in the last 20 years by using the statistical layers derived from the Hansen data that we now provide as part of our dataset.

**2. Response to Comments from Reviewer 2**

**The method has similarities to the biomass age curve approach (Chazdon et al., 2016; Zhang et al., 2017), but instead of using observed age curves, the random forest regression effectively uses climatological data to implicitly predict stand biomass growth curves, combining these with the biomass information to produce age. I.e. It assumes that growth and (non-stand-replacing) mortality rates can be estimated well by climatological variables alone. Given the importance of edaphic variables and management for determining these rates, I was a bit surprised not to see them included in the set of predictors. Whilst management is a tricky thing to assess globally, there are at least roughly related variables, e.g. human development index, which could have been tested. Possibly there is substantial autocorrelation of management with climate, which may help out here in terms of the accuracy of the classification, but then this would also potentially confound the current interpretation of climatic variables relating to Fig. 5. At the very least it would be good to see some discussion of the potential influence of management, if not testing the importance of some management and soil-related variables.**
**Response:**

Thank you very much for your comments. We now have included soil-related variables in our set of predictors collected from the HSWD dataset. However, none of the soil-related variables was selected during the feature selection procedure (see method section in the MS).

We acknowledge that management practices can modify the forest age-AGB relationship (i.e., growth and mortality rates) across different management regimes. Although there is no global product of management, we intended to create two proxies for management derived from the Hansen tree cover dataset:

- The intensity of tree loss from the Hansen tree cover loss layer (Hansen et al., 2010, Science). This metric was derived by counting the 30m pixels that experienced a tree cover loss for the period 2000-2019 within a 1km pixel.

- Last time since tree cover loss from Hansen tree cover loss layer (Hansen et al., 2010, Science) – standard deviation metric. This metric was calculated as the last time from 2019 since a 30m

pixel experienced tree cover loss, and we further computed the standard deviation of this last time since tree cover loss within 1km.

We acknowledge that these proxies can also be related to other types of disturbances than human-induced disturbances or management (e.g., tree cover loss due to drought). Still, they very likely contain relevant signals related to management and harvesting practices. For instance, we compared the two metrics derived from the Hansen data with a management regime product for Europe (Nabuurs et al., 2018), and there seems to be an association between the Hansen derived metrics and the management dataset.

Interestingly, these metrics were not selected during the feature selection procedure, and the final model still relies only on climatic data and biomass estimates.

The statistics derived from the Hansen data are now provided as part of our datasets whether users want them as proxies for disturbances regimes and applied some filtering in our forest age products (e.g., filtering out pixel that did not experience disturbances in the last 20 years). In addition, we added discussion points in our paper related to the role of management on the age-AGB relationships as well as the role of soil-related variables: *"Biomass estimates contain information about the current state of the forest, integrating the cumulative effect of land-use change, management and disturbance history. Having biomass (i.e., agb) as an important variable in predicting forest age suggested strong controls of management and disturbance regimes on the forest age distribution (ref)."*

**As things stand, I think that the interpretation of the influence of climate variables on the age estimates (Fig. 5) has to be made in the context of these variables effectively driving an implicit stand growth model which converts the biomass data into age. The biomass product is the only predictor which contains information about the current state of the forest including the cumulative effect of its full land-use, management and disturbance history. i.e. the climatic variables are not explaining age, they are (or at least likely are) explaining how biomass relates to age. A direct interpretation of climate effects on age would in any case be flawed because of management not being considered. I think that assessing the drivers of age distributions themselves (as opposed to trying to make the best map) would be clearer using a model which only included drivers and not also the state (i.e. biomass).**

**Response:**

Thanks for this comment. We agree that climate variables are explaining how age relates to biomass in our modelling framework. We have, therefore, adapted our discussion points on the link between climate and age in our manuscript: *"The importance of atmospheric water demand in explaining stand age variability could indicate how biomass is associated with stand*

*age across different climate regimes. More precisely, such observations could imply that high atmospheric water demand limits growth rates and maximum biomass, thereby indirectly controlling how biomass relates to age. In addition, high atmospheric water demand might influence fire frequency (Mueller et al., 2020) and indirectly control forest age distribution through the effect of fire on biomass."*

In addition, we agree that it would be very interesting to better understand the drivers of the age distribution globally by building a model that only uses its drivers while discarding state variables such as biomass. However, in this paper, we primarily intended to provide the best forest age map possible, a relevant dataset as pointed out by the referee. By including biomass as a predictor, we substantially increase the performance of the models and make sure that information related to land use, management, and disturbance history are implicitly considered through biomass estimates. Additionally, as the referee pointed out, it is challenging to retrieve global products that explicitly describe land-use, management and disturbance history.

**The masking out of low tree cover is a nice technique to reduce the negative bias on pixel biomass, and thus stand age, caused by a mix of forest and non-forest land covers within a 1 km2 pixel. However, although the size of a stand is a loosely defined concept, it is generally much smaller than 1 km2 . This leaves me wondering why the authors aggregate the biomass dataset from 100 m to 1 km? Is this to reduce noise? 100 x 100 m is already much bigger than most of the inventory plots which will underlie this product, but it would make the scale mismatch between plot-level training data and the biomass product used for extrapolation much less acute. Is a reduction in noise worth the loss of this important small-scale variation? An assumption of homogeneity within a 1 km2 cell (even after masking for tree cover) would tend to reduce the extremes of low and high biomass which would have been seen in the plot-level training data. This might go some way to explaining the relative dearth of young stands in the global age product (Figs. 9 and 10).**

**Response:**

Thanks for your comment. The main reason for aggregating the biomass dataset from 100m to 1km is to match the spatial resolution of the climate data (i.e., Worldclim data) for the upscaling procedure. We agree that by aggregating the biomass dataset from 100 m to 1km, we lose spatial variability as mentioned by the referee. Although we do not intend to provide a forest age product at 100m resolution, we added some discussion points about the limitation of having a global age product at 1km instead of, for instance, 100m pixel size: *"Finally, we assumed forest homogeneity within a 1 km grid-cell, which would reduce the extremes of low and high biomass estimates in the gridded global products that the models have learned in the plot-level training data. This limitation might, for instance, explain the relative dearth of very young stands (1-10 years old) in the MPI-BGC global age product (Figs.6A)."*

**Do all training datasets resolve up to 300 years of age (Line 100)? How did you deal with this if not? Are all the methods used to determine age likely to be accurate going back this far? Given the biases in prediction at greater than ca. 130 years (Fig., 3), and that the age of old-growth forest is both perspective and biome dependent, perhaps the classification between old-growth and non-old-growth would have been more accurate if a younger age threshold was taken? Whether or not you want to test this is of course up to you (perhaps you did already?), but at least a bit more clarity and discussion would be good.**

**Response:**

Thanks for this comment. Not all in-situ data resolve up to 300 years old as some of the old-growth forests are older than 500 years old, while sometimes the age of the plot is not even known (e.g., old-growth tropical forests). In our analysis, we used an arbitrary upper limit of 300 years old, which we agree can be discussed. However, we did additional experiments with an upper age limit of 150, and we observed a decrease in the model performance and a strong bias for the intermediate age class (>70 years old) (i.e., underestimating this age class). As mentioned by the referee, there are biases in prediction at greater than ca. 150 years; that is why we advised the user to use an upper age limit of 150 years old when using the MPI-BGC age product.

**I'm struggling with the interpretation around Fig. 7, where my reading of the graphs doesn't find the same features.**

- **L312. My reading of this graph is that in warm regions the whole range of forest ages is found, whereas in cooler regions, only a relatively narrow range of ages exists. This itself is pretty weird, as the plot seems to suggest that cold (presumably boreal) forests only have ages of around 100 years, with no younger stands. This doesn't seem very plausible.**
- **L320. It seems very strange that the youngest stands should only be found in the tropics (i.e. regions of high temperature, Fig. 7c). How can you explain this result?**
- **L323. I don't see any association between age and water availability in Fig. 7d. The points basically seem to form a square, apart from a small indent in the upper left, which seems challenging to interpret causally.**

**Response:**

Thank you for your comment. It is important to note that this plot has been done using a forest age product aggregated at a 0.25-degree pixel size; therefore, one loses resolution in the age spectrum (i.e., young and old forest age estimates were averaged at a 0.25-degree pixel size). This could explain the low fraction of very young forests in some regions mentioned by the

referee. We agree on the interpretation of the referee, in any case. Note that air temperature variables represent annual means. As mentioned by the referee, it appears that hot and dry regions have a substantial fraction of young forests while very cold regions (< 0 degC) have mainly old forests. Nevertheless, we still observe young stands (i.e., 10-20 years old) in relatively cold regions that correspond to boreal regions (e.g., annual mean around 0-5 degC).

**The comparison against other age products is very nice, however, not all comparator datasets are created equally. For instance, Pan et al. (2011) is based on a spatially systematic inventory system (at least in the US). Similarly, much of the temperate and boreal data in GFAD is based on summaries of statistics from national forest inventories. Whilst these come with substantial uncertainties, I would argue that they provide a sterner evaluation of the MPI-BGC product than the comparison with products based on biomass age curves. I suggest that this is reflected in the discussion of these results and also that the comparison with GFAD provides separate histograms for regions where GFAD is based on inventories and those where it is based on the biomass-age approach.**

**Response:**

Thanks for the advice. We agree that a direct comparison between the MPI-BGC product and other independent datasets is not systematically fair. We have followed the feedback from the referee by adding some discussion points in the manuscript as well as provided separate histograms for regions where GFAD is based on inventories and those where it is based on the biomass-age approach in Figure 10.

**Line 59. "Yet, an intrinsic..." It's unclear (at least to me) what this sentence is saying. Please can you rephrase it?**

**Response:**

Thanks for the comments. We have rephrased this sentence: "*Yet, stand age distribution can be modified to varying degrees of changes in environmental conditions and disturbance, therefore slowly change along with a forest age or successional continuum (Irvine et al., 2005; Piponiot et al., 2018).*"

**L125. Typo, "was covered in"**

**Response:**

Thanks for pointing this out. It has been corrected.

**L230. I don't think this result necessarily implies that vapr is a strong determinant of forest age. It can also imply that vapr is a strong control of how AGB relates to age. Given that high vapr is likely to limit both growth rates and maximum biomass, this seems to me the most likely interpretation. An influence of vapr on fire frequency, as suggested in the text, would surely primarily act through the effect of fire on AGB.**

**Response:**

We agree on the referee's interpretation of the vapr role in the age-AGB relationship. We have adapted our interpretation in the manuscript based on the referee's comment: "*The importance of atmospheric water demand in explaining stand age variability could indicate how biomass is associated with stand age across different climate regimes. More precisely, such observations could imply that high atmospheric water demand limits growth rates and maximum biomass, thereby indirectly controlling how biomass relates to age. In addition, high atmospheric water demand might influence fire frequency (Mueller et al., 2020) and indirectly control forest age distribution through the effect of fire on biomass.*"

**Fig. 3d. Are the under and overestimations distributed evenly geographically, or concentrated in specific regions? Providing some maps which break down the bias would be very useful for interpreting the results of the upscaled map.**

**Response:**

Thanks for this advice. We provided a map showing the residuals estimates at the plot level in figure 3.

**L288. Ceccherini et al. mainly show increases in Sweden and Finland. These increases are disputed (Palahi et al., 2021; Wernick et al., 2021) but also only occur after 2016 and therefore are not relevant to this map dated 2010. L290. See also (Vilén et al., 2012) for data on European age class distributions.**

**Response:**

Thanks for the comment. We agree that there is a temporal mismatch between our product and Ceccherini et al. study. We removed this citation from the manuscript and used the Vilén et al. study to discuss the age pattern in Europe.

**L292. I agree that different fire regimes seem plausible (Shorohova et al., 2011), but there is also substantial harvest identified in southern Siberia (Curtis et al., 2018), which should also play into this discussion.**

**Response:**

Thanks for the comment. We also added points related to harvesting when discussing the patterns in Siberia: "*The region of Siberia revealed a gradient of younger to older forests going from the South to the North part of the Siberian region (Fig. 6F). Such an observation could suggest different fire regimes between Southern and Northern Siberia (Shorohova et al., 2011) and confirm harvesting practices identified in Southern Siberia (Curtis et al., 2018).*"

**Fig. 4 caption. Clarification. "... in predicting forest age estimates in the regression model."**

**Response:**

Thanks for pointing this out. It has been clarified in the caption.

**L367-368. This sentence is really hard to follow, please can you rephrase it? I think what is being presented is the fraction of the random forest ensemble which predicts an old-growth forest for pixels which the mean of the ensemble attributes to old-growth, but after reading it several times I'm not 100% sure.**

**Response:**

Thanks. This sentence has been rephrased to improve clarity: "*First, we observed that the RFclassifier model had very high probabilities of classifying either a non-old-growth or an old-growth forest at pixel level as the fraction of the random forest ensemble to classify the two forest classes was generally close to one. (Fig. S1 and Fig. S2), suggesting relatively high confidence in the partitioning between old-growth and non-old-growth forests in the MPI-BGC forest age product.*"

**L392. Are you able to speculate a bit more on the methodological differences that drive the differences of the new forest age product with Pan et al.?**

**Response:**

We have added more discussion points in the manuscript explaining the methodological differences between the MPI-BGC and Pan et al products: "*Methodological differences between the Pan et al. (2011) and the MPI-BGC forest age datasets could explain such differences. While*

*the Pan et al. (2011) dataset integrate forest inventories, disturbance datasets, and land-use/land cover change data to retrieve forest age estimates in the Pan et al. (2011) dataset, the MPI-BGC forest age product relied on forest inventory,climate data and statistical methods. Additionally, forest inventory plots used to derive the MPI-BGC forest age product were relatively sparse in Canada (Fig. 1), which might limit the statistical methods used for the MPI-BGC forest age product to predict realistic forest age estimates (i.e., extrapolation issues). The fact that the Pan et al. (2011) dataset relies mainly on disturbances regimes inferred from optical remote sensing data (not biomass estimates) might explain the relatively higher fraction of young forests in the the Pan et al. (2011) dataset compared to the MPI-BGC dataset."*

**Figure 9. I don't follow why the age map without using a tree cover threshold needs to be applied for consistency. Surely the tree cover threshold deemed to be most appropriate should be used for comparison. Or, perhaps better, all tree cover thresholds should be compared and the one which provides the best agreement with the independent, inventory based, datasets (at least for regions where the inventory is systematic) might be recommended? It's really helpful that you explore this uncertainty due to tree cover and make all the maps available (thanks!), but many users will need to make a choice on a single map to use and it would be helpful to have a best recommendation to support this.**

**Response:**

Thanks for this feedback. The Chazdon age map product uses a biomass product for which there was no tree cover correction applied. That is why we thought it was fairer to compare the two products using the MPI-BGC age product without tree cover correction. However, we agree that it is relevant to compare all tree cover thresholds with the independent datasets to understand the one which provides the best agreement. We have added such a comparison in the manuscript.

**Figures 6 and 9. Is this mean or median age per pixel? If so, how did you deal with the old growth age class having an infinite upper age bound?**

**Response:**

In Figures 6 and 9, the age per pixel does not represent mean or median estimates but the age estimates that were predicted by the two random forests (i.e. the classifier and the regressor) at 1km spatial resolution. For the pixels classified as old-growth forests (i.e., having an infinite upper age bound), we assigned an age estimate of 300 years-old as described in the method section.

**Figure 10. The integrated forest area does not appear to be the same for GFAD and MPI-BGC products. Differences also appear to be shown in regions that have no forest (high northern latitudes, interior Australia). Please could you check? Also, which tree-cover correction was used in this comparison?**

**Response:**

Thanks for noticing this. We have now double-checked a potential mismatch in the forest area between the two products and corrected it. Here, we did not apply any tree correction in the MPI-BGC product as none was applied for the GFAD product. This was done to have a fair comparison between the two products.

---

## Author Response (AR2)

**Authors' Response to the Review Comments**

**Journal:** Earth System Science Data

**Manuscript #:** essd-2021-77

**Title of Paper: Mapping global forest age from forest inventories, biomass and climate data**

**Authors:** Simon Besnard, Sujan Koirala, Maurizio Santoro, Ulrich Weber, Jacob Nelson, Jonas Gütter, Bruno Herault, Justin Kassi, Anny N'Guessan, Christopher Neigh, Benjamin Poulter, Tao Zhang, Nuno Carvalhais

**Date Sent:** September 29th, 2021

We appreciate the time and efforts of the editor in reviewing this manuscript and the valuable suggestions offered. In addressing all issues indicated in the review report we trust that the revised version meets the Reviewers' comments and the journal's publication requirements.

**1. Response to Comments from topical editor**

**In the introductory text, please avoid repeating ''terrestrial in every line 42-44, pg.2. For istance, the text would flow equally well by deleting ''terrestrial'' in line 43 and 44. The context of the sentence makes it clear.**

**Response:**

Thank you for the suggestions. We have rephrased the two sentences as follows: "*Forests cover about 30% of the terrestrial surface of our planet and store a large part of the carbon, indicating their fundamental role in the carbon cycle (Bar-On et al., 2018). However, drivers controlling the capacity of the terrestrial biosphere to sequester carbon remain poorly characterized, limiting our understanding of the global land carbon sink's location (Cook-Patton et al., 2020).*"

**Supplementary-- Table S1. Isn't there a need of a time unit for ''time since last disturbance''?**

**Response:**

Thank you for noticing this. The metric used here year. We have changed it in Table S1.